# Longitudinal Changes in Dietary Supplement Use among United States Military Personnel: The US Military Dietary Supplement Use Study

**DOI:** 10.3390/nu16152547

**Published:** 2024-08-03

**Authors:** Joseph J. Knapik, Daniel W. Trone, Ryan A. Steelman, Emily K. Farina, Harris R. Lieberman

**Affiliations:** 1Military Nutrition Division, US Army Research, Institute of Environmental Medicine, 10 General Greene Ave., Natick, MA 01760, USA; emily.k.farina.civ@mail.mil (E.K.F.); harris.r.liberman.civ@health.mil (H.R.L.); 2Deployment Health Research Department, Naval Health Research Center, Building 329 Ryne Rd., San Diego, CA 92152, USA; daniel.w.trone.civ@health.mil; 3Defense Centers for Public Health–Aberdeen, 8252 Blackhawk Rd., Aberdeen Proving Ground, Gunpowder, MD 21010, USA; ryan.a.steelman.civ@health.mil

**Keywords:** vitamins, minerals, proteins, amino acids, herbals, joint health products, combination products, prohormones

## Abstract

Previous studies investigating temporal changes in dietary supplement (DS) use have used representative samples but have not followed the same cohort over time. This study investigated longitudinal patterns of changes in DS use and factors associated with discontinuing DS use in a single group of active-duty United States military service members (SMs). SMs (n = 5778) completed two identical questionnaires on their DS use and demographic/lifestyle characteristics an average ± standard deviation 1.3 ± 0.2 years apart. Prevalences of reported DS use ≥1 times/week in the baseline (BL) and follow-up (FU) phases were: any DS, BL = 77%, FU = 78%; multivitamins/multiminerals (MVM), BL = 50%, FU = 48%; individual vitamins/minerals, BL = 33%, FU 35%; proteins/amino acids, BL = 43%, FU = 39%; combination products, BL = 44%, FU = 37%; prohormones, BL = 5%, FU = 4%; herbal products, BL = 23%, FU = 21%; joint health products, BL = 12%, FU = 12%; fish oils, BL = 25%, FU = 23%; other DSs, BL = 17%, FU = 17%. Among BL users, the proportions reporting use in the FU phase were: any DS 88%, MVM 74%, protein/amino acids 70%, individual vitamin/minerals 62%, combination products 62%, fish oils 61%, joint health products 57%, herbal products 50%, other DSs 50%, and prohormones 37%. Higher odds of discontinuing any DS use in the follow-up were associated with female gender, younger age, higher BMI, and less weekly resistance training. Overall, prevalence of DS use was relatively consistent in the two phases; however, the cohort changed their use patterns in the follow-up with some discontinuing use and others initiating use, thus maintaining use prevalence over the period. These findings have implications for repeated cross-sectional DS studies where different samples are followed over time.

## 1. Introduction

Dietary supplements (DSs) are commercially available products consumed as an addition to the usual diet and include vitamins, minerals, herbs (botanicals), amino acids (AAs), and a variety of other products [1]. Marketing claims for some DSs include improvements in overall health status, enhancement of cognitive or physical performance, increases in energy, loss of excess weight, attenuation of pain, and other favorable effects. The regulatory framework for DS sales in the United States (US) was established by Dietary Supplement Health and Education Act of 1994 (DSHEA) [1]. The DSHEA limits the ability of the US Food and Drug Administration (FDA) to regulate DSs. Prior to bringing a new DS onto the market, manufacturers must notify the FDA 75 days in advance, but FDA approval is not required for retailing a DS product. Since the DSHEA became law, US sales of dietary supplements increased from USD 4 billion in 1994 to USD 60 billion in 2021 [2,3], a 15-fold increase over 28 years. Sales are expected to increase another USD 10 billion through 2025 [3]. Vitamin sales are the largest single DS category, accounting for over 30% of the market, but herbals (botanicals) are rapidly gaining market share, expected to represent 22% of sales by 2025 [3]. Protein/AAs are expected to have the largest compound annual growth rate at 7% from 2023 to 2030 [4].

In consonance with the increase in DS sales, the prevalence of DS use has been increasing over time. Some of the earliest studies (1970s) using data from the National Health and Nutrition Examination Survey (NHANES) revealed that 23% of adults used DSs [5]. More recent NHANES data (2019) indicated that about 57% of US adults use one or more DSs [6,7]. In 2006–2007, the prevalence of DS use in the Army was 53% [8] and the most current data (2019–2020) indicate that 76% of Army personnel use DSs [9]. Military personnel also have a higher use prevalence of DSs that are unique such as proteins, amino acids, and combination products [9] that are more often used by athletes [10].

Several civilian studies [7,11,12] and at least one military study [13] have examined longitudinal changes in DS use prevalence over time. However, these were between-subject studies that used different, but representative, cohorts to examine temporal trends, and we are not aware of any study that has followed the same cohort over time. Tracking DS use in the same cohort over time allows an examination of changes in patterns of DS use in individual participants. Thus, the purpose of the present study was to examine changes in temporal trends and patterns of DS use in a single cohort of United States (US) military service members (SMs). We hypothesized that the pattern of DS use would change over time such that some participants would continue use, some discontinue use, and others become new users. A secondary purpose was to examine factors associated with discontinuing DS use during a follow-up period.

## 2. Methods

This longitudinal investigation involved a survey completed twice by a stratified random sample of US active-duty military SMs. The Naval Health Research Center (NHRC) Institutional Review Board approved the investigation, and participants signed an informed consent document. Investigators adhered to policies and procedures for the protection of human subjects as prescribed by the Department of Defense Instruction 3216.01, and the research was conducted in adherence to provisions of the Title 32 Code of Federal Regulations, Part 219.

### 2.1. Sampling Frame and Solicitation Procedures

There were two phases of this study, baseline (BL) and follow-up (FU). Details of the sampling frame, solicitation of SMs, subject recruitment flow chart, and response bias for the BL phase have been previously reported [14]. Briefly, investigators requested from the Defense Manpower Data Center (DMDC) a random sample of 200,000 SMs stratified by sex (88% male and 12% female) and branch of service (Army 36%, Air Force 24%, Marines 15%, and Navy 25%). Additional data obtained from DMDC on the random sample included date of birth, sex, rank, time in service, and educational level. Recruitment of the randomly selected SMs into the BL study involved a maximum of 12 sequential contacts. These included an introductory postal letter, a follow-up email message after 10 days, a postcard three weeks later, and up to seven emails and three post card reminders evenly distributed across the time the survey was open. After this, contact with the SM ended. All postal and on-line contacts stated that at any time the SM could decline participation and be removed from the contact list. Recruitment into the BL phase began in December 2018 and ended in August 2019.

As part of the BL informed consent, potential participants were informed there would be an FU phase that would involve the same procedures. Prior to the FU phase, DMDC identified SMs no longer on active duty so they would not receive an FU request. Other SMs who volunteered for the BL phase and were still on active duty were asked to participate in the FU phase in a letter sent about 9 months after the BL phase completion. Solicitation procedures were the same as in the FU phase with 12 sequential contacts. Recruitment into the FU phase began in April 2020 and ended in December 2020.

### 2.2. Survey Description

An identical survey was used in the BL and FU phases, and it was similar to instruments previously used to examine DS use in military DS investigation [15]. It was completed by participants online. The survey was designed to describe participants and obtain types and frequencies of DSs used. To characterize participants, there were questions on demographics (gender, age, education level, height, weight, service branch) and lifestyle (amount of exercise, tobacco use, alcohol consumption). DS use questions included 96 generic DSs (e.g., multivitamins/multiminerals [MVMs], individual vitamins and minerals, proteins/AAs, herbals, joint health products, fish oils) and 91 brand name products. The brand name products included some of those used in previous armed forces studies [8,16,17], but individual items were updated based on a review of DS inventories in the Army, Navy, and Air Force Exchange Services and General Nutrition Center stores on or near military installations before the start of the BL phase. There were also open text fields on the questionnaire where SMs could include supplements not on the provided lists. For each listed DS, SMs were asked to estimate how frequently each supplement was used during the past 6 months (“never”, “once a month”, “once a week”, “2–6 times/week”, or “daily”). SMs were also asked to estimate how much money they spent on supplements monthly. Table 1 provides the DS category definitions used in this study.

### 2.3. Data Analyses

All statistical analyses were conducted using International Business Machine (IBM) SPSS Statistics, version 26 or 27 (IBM, Armonk, NY, USA). Body mass index (BMI) was computed from the questionnaire responses as weight/height^2^ (kg/m^2^). Weekly duration of aerobic and resistance training (minutes/week) was calculated by multiplying reported weekly exercise frequency (sessions/week) by reported duration of training (minutes/session). Alcohol consumption was quantified using the National Institute of Health assumption that a “standard drink” contained 17.74 mL of alcohol [19]. Standard drinks included 12 ounces of regular beer or fermented fruit drink (5% alcohol), 8.5 ounces of higher alcohol beer (7% alcohol), 5 ounces of wine (12% alcohol), 4.25 ounces of fortified wine (15% alcohol), and 1.5 ounces of liquor (40% alcohol). Supplements that service members placed in the “other” categories were examined, and responses were placed into their proper DS category or listed as “other” supplements if they did not fit a defined DS category (Table 1).

Demographic and lifestyle data in the BL and FU phases were compared using the Friedman Test for related samples [20]. For SMs reporting use ≥1 time/week, DS use prevalence (%) with 95% confidence interval (95%CI) were calculated for each DS category (Table 1) in the FU and BL phases. Changes in prevalence between the two periods were calculated. Two by two tables were constructed to examine continued DS use, continued non-use, and changes in use in each DS category (i.e., BL use/non-use x FU use/non-use). These four classifications included consistent users (BL user also reporting use in the FU), discontinuing users (BL users no longer reporting use in the FU), new users (BL non-users reporting use in the FU), and never users (SMs not reporting use in either phase). The McNemar test for repeated measures [21] was used to compare the BL and FU phases. The monthly amount of money spent on DSs in the BL and FU phases were examined using paired sample *t*-tests. The number of DSs reported in the BL and FU phases was calculated for each SM, and SMs were placed into one of four groups in each phase: 0 (no DS use), 1–2, 3–4, and ≥5 DSs. Data were graphed, and the McNemar test [21] was used to compare the proportion of SMs in the BL and FU phases in each of the four groups separately.

Multivariable logistic regressions were used to examine demographic and lifestyle factors independently associated with discontinuing DS use in each DS category. The dependent variable was “discontinuing use” (yes/no) and the independent variables included all the demographic and lifestyle factors (gender, age, formal education, BMI, Hispanic ethnicity, race, service branch, tobacco use [smoking and smokeless], alcohol use, and weekly duration of exercise [aerobic and resistance training]). For each independent variable, a reference stratum was selected (defined with an odds ratio = 1.00), and other strata were compared to the reference.

To address response bias, FU responders (n = 5778) were compared to non-responders/non-contacted SMs in BL or FU phases (n = 194,222). Chi-square statistics were used to compare the two groups by sex, military service, education level, and military rank. Independent sample t-tests assessed differences in age and time in service.

## 3. Results

Figure 1 presents the study flow chart of the SM recruitment, loss, and participation in the BL and FU phases of the study. From the initial sample frame of 200,000 service members, 73% (n = 146,365) were successfully contacted (i.e., no returned postal mail) at BL and of these, 26,680 (18.2%) signed the informed consent and completed the BL questionnaire. Of the 26,680 BL responders, 22,858 (86%) were still on active duty at the start of the FU phase and were successfully contacted at least once during FU phase. Of these, 5778 completed the FU questionnaire for an FU response rate of 25.3% (5778/22,858). The average ± SD follow-up time (time from BL to FU questionnaire completions) was 15.8 ± 2.0 months with a range of 9.9–22.8 months.

### 3.1. Demographic and Lifestyle Factors in Baseline and Follow-Up Phases

Table 2 compares the demographic and lifestyle factors of participants at BL and FU phases. There were no significant differences among the strata for reported gender, Hispanic ethnicity, race, aerobic exercise duration, or service branch. Compared to the BL phase, more participants in the FU phase were reported in the older age strata, higher formal education strata, and higher calculated BMI strata. There were significant differences between phases for tobacco use, with fewer SMs reporting smoking or smokeless tobacco use in the FU phase. The distribution of alcohol consumption redistributed compared to the BL phase, with more participants in the FU phase reporting being non-users as well as users in the highest consumption category. There were more participants reporting in the shorter duration strata for resistance exercise in the FU phase.

### 3.2. Changes in Prevalence and Patterns of Use by DS Category

Table 3 provides prevalence of DS use in the BL and FU phases. Overall prevalences were similar in the two phases for most DS categories. Use prevalence slightly increased in the FU phase for any DS and vitamins/minerals. Use prevalence slightly decreased in the FU phase for MVMs, proteins/AAs, combination products, prohormones, herbal products, fish oils, and other DSs. There was no change in use prevalence for joint health products.

Table 4 shows the changes in DS categories in the BL and FU phases. Among users at BL, 88% to 37% reported consistent use while 63% to 13% discontinued use in the FU phase. In descending order, categories with the highest prevalence of discontinued use were prohormones, other DSs, herbal products, joint health products, fish oils, combination products, individual vitamins/minerals, proteins/AAs, MVMs, and any DS. Among non-users at BL, 45% to 2% reported using in the FU period while 98% to 55% reported never using in either phase. In descending order, the categories with the largest proportion of new users in the FU were any DS, MVMs, vitamins/minerals, combination products, proteins/AAs, herbal products, other DSs, fish oils, joint health products, and prohormones.

Figure 2 shows the number of DSs reported by SMs in the BL and FU phases. Compared to the BL phase, fewer SMs reported use of ≥5 DSs and more SMs reported use of 1–2 DSs in the FU phase. The amount of money (mean ± SD) spent on DSs was USD 44 ± 60 per week at BL and USD 44 ± 59 per week at FU (*p* = 0.82). A total of 22% and 20% spent >USD 50 in the BL and FU phases, respectively.

### 3.3. Factors Associated with Discontinuing DS Use

Table 5 shows the multivariable analysis of the demographic and lifestyle factors associated with discontinuing DS use in each DS category. There are 10 models, one for each DS category with all the demographic and lifestyle factors included. Ninety-five percent of SMs (n = 5483) had complete data on all factors and were included in these analyses.

Table 5 shows that higher odds of discontinuing any DS were associated with female gender, younger age, higher BMI, and shorter weekly duration of resistance exercise. Higher odds of discontinuing MVM use were associated with female gender, being of multiple races, and longer duration of resistance training. Discontinuing individual vitamin/mineral use was associated with female gender, younger age, and longer duration of resistance training; moderate alcohol use was associated with lower odds of discontinuing individual vitamin/mineral use Discontinuing protein/AA use was associated with female gender, younger age, higher BMI, having quit smoking, high alcohol use, and longer duration of resistance training; those of other races were less likely to discontinue use compared to whites. Higher odds of discontinuing combination product use were associated with female gender, less formal education, and higher BMI. Higher odds of discontinuing prohormone use were associated with male gender, higher BMI, having quit smokeless tobacco use, longer duration of resistance training, and service in the Army (compared to the Air Force). Higher odds of discontinuing herbal products were associated with female gender, being of multiple races, and longer duration of resistance training. Higher odds of discontinuing joint health products were associated with female gender, older age, higher BMI, and longer duration of resistance training; quitting smoking was associated with lower odds of discontinuing joint health products. Higher odds of discontinuing fish oil use were associated with being of multiple races and longer duration of resistance training. Higher odds of discontinuing other DSs were associated with female gender, higher BMI, higher alcohol use, and longer duration of resistance training.

### 3.4. Response Bias

Compared to the requested stratified sample, the FU cohort were more likely to be female (12% vs. 14%, *p* < 0.01) and consisted of more Air Force personnel with fewer personnel from other services (Air Force 39%, Army 31%, Marine Corps 10%, Navy 19%, *p* < 0.01). Compared to those not participating in the larger sample frame (n = 194,222), the FU cohort at BL was older (29 ± 7 vs. 35 ± 8 yr, *p* < 0.01), had more time in service (8 ± 6 vs. 12 ± 7 yr, *p* < 0.01), achieved higher formal educational levels (21% vs. 59% with college degrees, *p* < 0.01), and were more likely to be officers (17% vs. 43%, *p* < 0.01).

## 4. Discussion

This investigation found that prevalence of DS use among SMs was relatively stable within DS categories during an average 1.3 years of follow-up. However, this concealed the fact that in many DS categories, the distribution of individual users and non-users changed from the BL to FU phase. Some SMs reported discontinuing DS use while others started use in the FU, thus maintaining the relatively stable overall prevalence in both phases for virtually all DS categories. For example, MVM use prevalence in the BL and FU phases was 50% and 48%, respectively. However, only 72% of BL users reported using DSs in both study phases, while 28% of BL users no longer reported use in the FU. The difference was largely compensated for by “new” users resulting in similar prevalences in both study phases. For many DS categories, factors independently associated with discontinuing DS use included female gender, higher BMI, and longer duration of resistance training.

### 4.1. Prevalence of DS Use

The prevalence of DS use in the FU cohort (n = 5778) was slightly higher than that of the larger BL cohort of this study (n = 26,681) previously published [14] (74% compared to 77%). Prevalence differences between the FU and larger BL cohorts were greatest for MVMs (BL cohort = 45%, FU cohort at BL = 50%) and smallest for combination products (44% in both cohorts at BL) and prohormones (5% in both cohorts at BL) [14]. The FU cohort appears somewhat more likely to be DS users in most categories compared to the larger BL cohort, although prevalence differences were relatively small.

Military personnel have a higher prevalence of DS use than the general US population. Recent data (2019) from NHANES indicated that 56% to 58% of Americans reported using DSs [6,7] while 74% of military personnel in 2019–2020 reported using DSs [9]. Compared to their civilian counterparts, SMs are more likely to use different types of DSs, especially proteins/AAs and combination products. For example, only 4% of an NHANES cohort reported amino acid use [12], while in the current study, 39% to 43% reported use of proteins/AAs. Caution must be used in these comparisons because of different methods of data collection. Recent NHANES studies asked participants to report on DS use in the past 30 days [6] or the past 30 days combined with two 24 h recalls [7], whereas in the present study, use ≥1 time/ week in the last 6 months was reported. Also, NHANES cohorts are older than the military cohorts, and older individuals have a higher DS use prevalence than younger ones [22,23].

Military personnel also have a higher DS use prevalence than many athletic groups. One comprehensive meta-analysis indicated that a weighted average of 60% of athletes reported DS use, but with considerable differences among sports [10]. For example, DS use was reported by 87% to 100% of bodybuilders [24,25,26], elite skiers [27], elite swimmers [28], elite rowers [29], professional soccer players [30], and other elite athletes [31,32]. About 43% of SMs reported use of protein/AAs in the BL phase, compared to a weighted average of 15% in the comprehensive meta-analysis of athletes’ DS use [10], but again there were major differences among sports.

### 4.2. Factors Associated with Discontinuing DS Use

The current study was unique as it investigated factors associated with discontinuing DS use over time, which no previous study had explored. We found that women were more likely to discontinue use in most DS categories compared to men. In previous studies, women were more likely than men to have a higher prevalence of DS use in most DS categories in both civilian [5,7,11,12,22,33,34,35,36,37,38] and military [8,14,16,17] investigations. Women are more likely than men to make behavioral changes to improve health [39,40,41], which may result in experimenting with DSs and possibly discontinuing use if DSs are not achieving desired effects.

Cross-sectional civilian studies have consistently shown that DS use increases with age [5,7,12,22,33,34,35,36,42], but investigations of military SMs have generally indicated the lowest overall DS use is in the youngest age group (18–24 years) with only modest increases in prevalence in older age groups. Use of protein/AA is unique in that use generally decreases with age among SMs [13,14,17,43]. The current data add that discontinuation of any DS and proteins/AA decreases with age, suggesting older SMs are more persistent in their overall use of DSs, especially for protein/AAs. On the other hand, previous studies show that use of joint health products increase with age [14,17], possibly associated with the higher incidence of osteoarthritis in older individuals [44] and data suggesting some efficacy for certain types of joint health products [45]. The current study indicated higher odds of discontinuation of joint health products as age increased, suggesting less persistent use in this DS category over time among older SMs.

Military studies have consistently reported that higher BMI was associated with higher DS use [8,14,17], in contrast with civilian studies [7,22,33,34,35,38,46], which have generally shown little to no relationship between BMI and DS use. The largest difference in use prevalence between non-obese (<25.0 kg/m^2^) and obese (>30.0 kg/m^2^) SMs have been for combination products and prohormones [16,17], categories that often contain substances marketed for weight or fat loss [47]. There are strict weight for height and body fat standards that SMs must meet to continue their military service [48,49,50,51]. Those who do not meet these criteria can receive adverse performance reports and be discharged from service. SMs who have difficulty meeting these requirements may be encouraged to use DSs marketed for weight or body fat loss. The current study adds that at higher BMI levels, SMs are also more likely to discontinue use in many DS categories including protein/AAs, combination products, prohormones, and joint health products, although the reasons for this cannot be determined given the data obtained.

As resistance training duration increased, SMs were less likely to discontinue use of any DS; however, in most other DS categories, as resistance training duration increased, SMs were more likely to discontinue use. This suggested that SMs performing more resistance training discontinued DS use in specific DS categories, switching to other categories, resulting in less overall discontinuing of DS use (i.e., any DS). Higher odds of discontinuing use as training duration increased was apparent in almost all DS categories apart from combination products where the odds of discontinuing use were similar across duration strata. Previous studies have shown that as resistance training increases, so does the use of combination products [8,14,17] and individuals doing more resistance training appear to be less likely to discontinue use in this category. Many combination products are promoted for their purported ergonomic effects and ability to increase muscle mass and strength [52,53], which could encourage more active SMs to consistently use them.

### 4.3. Monthly Resources Spent on DSs

Military studies conducted between 2006 and 2014 indicated that SMs reported spending an average of USD 38 to USD 39 per month on DS [8,16,17]. In the BL phase of this study conducted in 2019 [14], the larger BL cohort reported spending USD 40 per month on DSs with 31% spending >USD 50. Individuals participating in the FU phase reported spending USD 44 per month in both BL and FU phases, but only 20%–22% reported spending >USD 50 per month. Thus, although overall spending was greater in the FU cohort (compared to the larger BL cohort), there were fewer individuals spending larger amounts of money. Given potential cost inflation in the DS market, the small increase in monthly spending seen in our more recent studies, compared to the earlier studies [8,16,17], may not be surprising and may even be less than expected.

### 4.4. Strengths and Limitations

This study examined temporal changes in patterns of DS use in a large sample of SMs from all branches of service. The questionnaire used was identical in the BL and FU phases and based on questionnaires used in previous military studies [15], but updated for DSs currently available to SMs. The demographics and lifestyle factors examined were like those examined in other civilian and military investigations, which allowed for reasonable comparisons among studies. However, there were limitations. As indicated by the response bias, the service members who volunteered for the study differed from the desired stratified sample, although both sexes and all service branches were well represented. The 25% response rate to the second questionnaire was relatively low, but close to the 20% response rate expected for military survey studies conducted by NHRC that investigated topics other than DSs. Although the questionnaire instructions and consent form emphasized the importance of both DS user and non-user participation, it is possible the FU questionnaire may have attracted slightly more DS users. This was suggested by the slightly higher overall DS prevalence and the larger amount of money spent on DSs monthly by the FU cohort, compared to the larger BL cohort. All data were self-reported and had the usual weaknesses associated with this method, including recall bias, social desirability, errors in self-observation, and inadequate recall [54,55].

## 5. Conclusions

This investigation indicated that a cohort of SMs had only minor changes in the prevalence of use in DS categories during an approximate 1-year period. However, this relatively stable prevalence was maintained not only by consistent users, but also by new users who largely compensated for those who discontinued use within the FU phase. MVMs and proteins/AAs had the largest proportion of consistent users while the prohormones category was where SMs were most likely to discontinue use. It would be useful to conduct similar studies in civilian populations for comparison to the military, especially since SMs use considerably different types of DSs compared to civilians, most notably proteins/AAs and combination products. This paper provides basic information on the temporal patterns of DS use by military personnel and how demographic and lifestyle factors are associated with discontinuing DS use.

## Figures and Tables

**Figure 1 nutrients-16-02547-f001:**
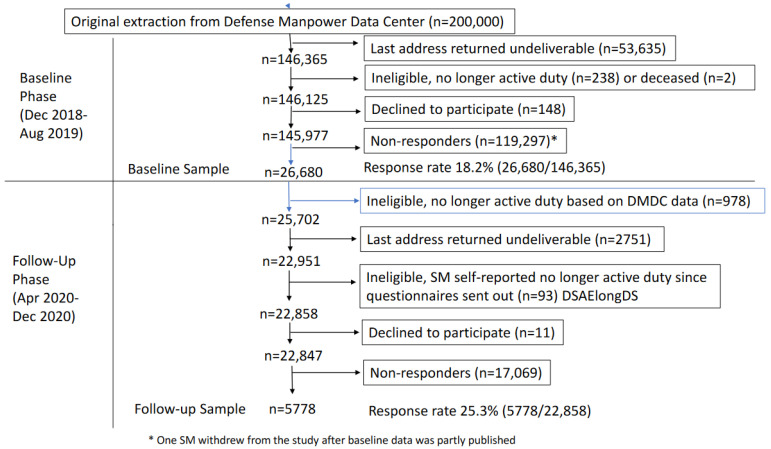
Flow chart summarizing service member recruitment, loss, and participation in baseline and follow-up phases.

**Figure 2 nutrients-16-02547-f002:**
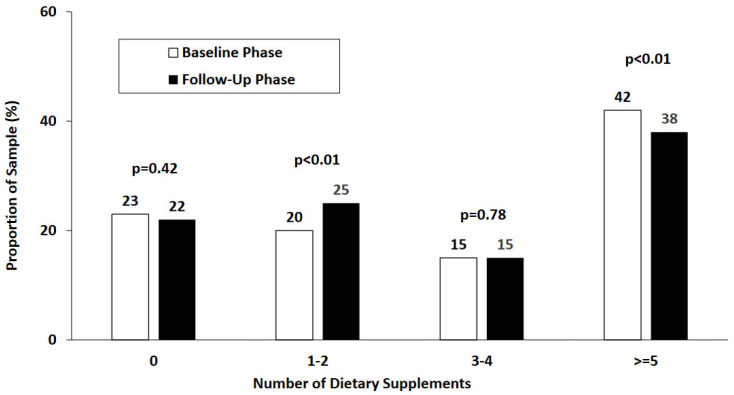
Number of dietary supplements reported by service members in baseline and follow-up phases.

**Table 1 nutrients-16-02547-t001:** Dietary supplement categories in study of US service members.

Category	Definition
Dietary Supplement	Any substance defined by DSHEA
Multivitamin/Multimineral	DS containing two or more vitamins and/or two or more minerals with no additional supplement ingredients
Individual Vitamin or Mineral	DS that is a single vitamin or mineral supplement, such as vitamin D or calcium
Protein or Amino Acid	AA mixtures, protein powders, and similar products where the intent is to provide a single AA or complex protein source
Combination Product	DS with mixtures of ingredients from any of the above categories, including two or more categories and multiple ingredients. Includes products marketed as weight loss, pre- or post-workout supplements, and muscle-/body-building products.
Purported Prohormone	Steroidal hormone or herbal substitute for hormones marketed as a DS and included on the Nutrition and Supplement Facts label.
Herbal	DS that includes one or more herbal ingredients with no nutrient or other supplement ingredient. Includes plant-derived ingredients.
Joint Health Product	Substance that purports to improve the functioning of body joints, such as glucosamine (with or without chondroitin) or methylsulfonylmethane
Fish Oil	Substances derived from the tissues of oily fish and presumably containing omega-3 fatty acids
Other Dietary Supplement	Other DSs that do not fit into the categories above

Abbreviations: AA, amino acid; DS, dietary supplement; DSHEA, Dietary Supplement Health and Education Act of 1994 [18].

**Table 2 nutrients-16-02547-t002:** Comparison of demographic and lifestyle factors of follow-up cohort in baseline and follow-up phases.

Variable	Strata	Baseline		Follow-Up		*p*-Value (Friedman Test)
		Sample Size (n)	Proportion of Sample (%)	Sample Size (n)	Proportion of Sample (%)	
All SMs	None	5778	100.0	5778	100.0	-----
Gender	Men	4972	86.1	4972	86.1	-----
	Women	806	13.9	806	13.9	
Age	18–24 yr	581	10.1	372	6.4	<0.01
	25–29 yr	948	16.4	873	15.1	
	30–39 yr	2620	45.7	2623	45.5	
	≥40 yr	1578	27.6	1902	33.0	
Formal Education	Some HS/HS Grad	438	7.6	346	6.0	<0.01
	Some College	1962	34.0	1895	33.0	
	Bachelor/Grad Deg	3378	58.5	3502	61.0	
Body Mass Index	<25.0 kg/m^2^	1758	31.0	1643	28.8	<0.01
	25.0–29.9 kg/m^2^	3094	54.5	3138	55.0	
	≥30.0 kg/m^2^	827	14.6	922	16.2	
Ethnicity	Not Hispanic	5057	87.5	5032	87.7	0.21
	Hispanic	721	12.5	706	12.3	
Race	White	4515	78.1	4429	76.7	0.12
	Black	491	8.5	477	8.3	
	American Indian	37	0.6	38	0.7	
	Asian	245	4.2	240	4.2	
	Pacific Islander	28	0.5	25	0.4	
	Other	159	2.8	165	2.9	
	Multiple	303	5.2	356	6.2	
Smoking	Never Smoked	3851	67.5	3744	69.0	<0.01
	Smoked but Quit	1023	17.9	980	18.1	
	Current Smoker	833	14.6	699	12.9	
Smokeless Tobacco	Never Used	4640	82.5	4465	82.9	0.01
	Used but Quit	452	8.0	466	8.6	
	Current User	532	9.5	458	8.5	
Alcohol Consumption	0 mL/day	1498	25.9	1664	28.8	<0.01
	0.23–24.85 mL/day	1404	24.3	1048	18.1	
	24.86–71.69 mL/day	1400	24.2	1269	22.0	
	>71.69 mL/day	1475	25.5	1797	31.1	
Aerobic Exercise	≤90 min/wk	1560	27.0	1719	29.8	0.16
	91–180 min/wk	1685	29.2	1570	27.2	
	181–300 min/wk	1325	22.9	1310	22.7	
	>300 min/wk	1208	20.9	1179	20.4	
Resistance Exercise	<45 min/wk	1775	30.7	2556	44.2	<0.01
	46–135 min/wk	1466	25.4	1313	22.7	
	136–300 min/wk	1415	24.5	1205	20.9	
	>300 min/wk	1122	19.4	704	12.2	
Service Branch	Air Force	2295	39.7	2296	39.7	0.64
	Army	1807	31.3	1805	31.2	
	Marine Corps	574	9.9	574	9.9	
	Navy	1102	19.1	1103	19.1	

Abbreviations: HS = high school, Grad = graduate.

**Table 3 nutrients-16-02547-t003:** Prevalence of DS use (≥1 time/week) among service members in baseline and follow-up phases.

DS Category	Baseline		Follow-Up		Change (%) ^a^	Prevalence Ratio (Follow-Up/Baseline)
	n	Prevalence % (95%CI)	n	Prevalence % (95%CI)		
Any DS	4467	77.3 (76.1–78.5)	4494	77.8 (76.6–79.0)	0.5	1.01
MVMs	2884	49.9 (48.1–51.7)	2791	48.3 (46.8–49.8)	−1.6	0.97
Vitamins/Minerals	1912	33.1 (31.0–35.2)	2017	34.9 (32.8–37.0)	1.8	1.06
Proteins/Amino Acids	2486	43.0 (41.1–44.9)	2261	39.1 (37.1–41.1)	−3.9	0.91
Combination Products	2534	43.9 (42.0–45.8)	2144	37.1 (35.1–39.1)	−6.8	0.84
Prohormones	304	5.3 (2.8–7.8)	233	4.0 (1.5–6.5)	−1.3	0.76
Herbal Products	1339	23.2 (20.9–25.5)	1208	20.9 (18.6–23.2)	−2.3	0.90
Joint Health Products	693	12.0 (9.6–14.4)	695	12.0 (9.6–14.4)	0.0	1.00
Fish Oils	1466	25.4 (23.2–27.6)	1327	23.0 (20.7–25.3)	−2.4	0.91
Other DSs	1007	17.4 (15.1–19.7)	994	17.2 (14.9–19.5)	−0.2	0.99

^a^ Calculated as follow-up prevalence minus baseline prevalence. Abbreviations: DS = dietary supplement, MVM = multivitamin/multimineral, 95%CI = 95% confidence interval.

**Table 4 nutrients-16-02547-t004:** Changes in dietary supplement use (≥1 time/week) by service members at follow-up phase.

DS Category	Baseline Users				Baseline Non-Users				McNemar Test *p*-Value
	User at Baseline Reporting Use at Follow-Up (Consistent Users)		Users at Baseline No Longer Reporting Use at Follow-Up (Discontinued Users)		Non-Users at Baseline Reporting Use at Follow-Up (New Users)		Non-Users at Both Baseline and Follow-Up (Never Users)		
	n	Prevalence (%)	n	Prevalence (%)	n	Prevalence (%)	n	Prevalence (%)	
Any DS	3908	87.5	559	12.5	586	44.7	725	55.3	0.44
MVMs	2061	71.5	823	28.5	730	25.2	2164	74.8	0.02
Vitamins/Minerals	1188	62.1	724	37.9	829	21.4	3037	78.6	<0.01
Protein/Amino Acids	1749	70.4	737	29.6	512	15.6	2780	84.4	<0.01
Combination Products	1572	62.0	962	38.0	572	17.6	2672	82.4	<0.01
Prohormones	112	36.8	192	63.2	121	2.2	5353	97.8	<0.01
Herbal Products	670	50.0	669	50.0	538	12.1	3901	87.9	<0.01
Joint Health Products	392	56.6	301	43.4	303	6.0	4782	94.0	0.97
Fish Oils	892	60.8	574	39.2	435	10.1	3877	89.9	<0.01
Other DSs	498	49.5	509	50.5	496	10.4	4275	89.6	0.71

Abbreviations: DS = dietary supplement, MVM = multivitamin/multimineral.

**Table 5 nutrients-16-02547-t005:** Multivariable associations between discontinuation of dietary supplement use and demographic/lifestyle factors among service members over 1.3 ± 0.2 years ^a^.

Variable	Strata	Any Dietary Supplement	Multivitamin/Multiminerial	Vitamin/ Mineral	Protein/Amino Acid	Combination Product	Prohormone	Herbal	Joint Health Product	Fish Oil	Other Dietary Supplements
Gender	Male	1.00	1.00	1.00	1.00	1.00	1.00	1.00	1.00	1.00	1.00
	Female	1.31 (1.02–1.60)	1.34 (1.07–1.67)	1.49 (1.19–1.87)	1.29 (1.01–1.64)	1.35 (1.09–1.67)	0.21 (0.08–0.59)	1.85 (1.47–2.33)	1.65 (1.15–2.35)	1.04 (0.78–1.39)	1.76 (1.35–2.30)
Age	18–24 yr	1.00	1.00	1.00	1.00	1.00	1.00	1.00	1.00	1.00	1.00
	25–29 yr	0.84 (0.60–1.18)	1.03 (0.75–1.42)	1.04 (0.75–1.44)	0.84 (0.62–1.15)	1.05 (0.78–1.41)	1.03 (0.55–1.93)	1.24 (0.88–1.73)	1.41 (0.71–2.83)	1.15 (0.80–1.66)	0.95 (0.64–1.40)
	30–39 yr	0.61 (0.44–0.84)	1.14 (0.85–1.52)	1.10 (0.81–1.48)	0.86 (0.64–1.15)	1.00 (0.76–1.32)	1.28 (0.72–2.28)	1.01 (0.73–1.39)	2.97 (1.60–5.52)	1.05 (0.74–1.48)	0.97 (0.68–1.39)
	≥40 yr	0.51 (0.36–0.74)	1.06 (0.76–1.47)	0.89 (0.63–1.26)	0.63 (0.45–0.89)	1.02 (0.75–1.39)	1.27 (0.66–2.42)	1.07 (0.75–1.53)	5.09 (2.66–9.76)	1.15 (0.78–1.69)	1.11 (0.74–1.64)
Formal Education	Some HS/HS Grad	1.00	1.00	1.00	1.00	1.00	1.00	1.00	1.00	1.00	1.00
	Some College/AA	0.79 (0.56–1.12)	0.87 (0.63–1.19)	1.12 (0.79–1.57)	0.82 (0.60–1.13)	0.76 (0.57–1.01)	1.42 (0.76–2.66)	1.26 (0.87–1.84)	1.48 (0.77–2.83)	1.07 (0.73–1.57)	1.14 (0.75–1.75)
	College Grad	0.79 (0.55–1.13)	0.84 (0.61–1.16)	1.06 (0.74–1.51)	0.83 (0.59–1.15)	0.69 (0.51–0.93)	1.37 (0.71–2.64)	1.12 (0.76–1.66)	1.22 (0.63–2.37)	1.15 (0.78–1.71)	1.20 (0.77–1.86)
Body Mass Index	<25.0 kg/m^2^	1.00	1.00	1.00	1.00	1.00	1.00	1.00	1.00	1.00	1.00
	25.0–29.9 kg/m^2^	1.18 (0.96–1.46)	1.02 (0.85–1.22)	1.10 (0.91–1.32)	1.01 (0.83–1.22)	1.11 (0.93–1.31)	1.55 (1.02–2.36)	1.11 (0.91–1.36)	1.13 (0.83–1.53)	1.19 (0.96–1.48)	1.18 (0.94–1.49)
	≥30 kg/m2	1.37 (1.02–1.84)	0.86 (0.66–1.11)	1.19 (0.91–1.54)	1.39 (1.08–1.79)	1.45 (1.15–1.82)	2.20 (1.33–3.62)	1.28 (0.98–1.68)	1.60 (1.10–2.33)	1.31 (0.97–1.75)	1.36 (1.00–1.85)
Hispanic Ethnicity	No	1.00	1.00	1.00	1.00	1.00	1.00	1.00	1.00	1.00	1.00
	Yes	0.82 (0.61–1.11)	0.89 (0.70–1.14)	1.15 (0.90–1.47)	1.20 (0.94–1.54)	1.11 (0.89–1.39)	1.22 (0.80–1.87)	1.10 (0.85–1.42)	0.97 (0.66–1.42)	1.11 (0.84–1.44)	1.11 (0.83–1.47)
Race	White	1.00	1.00	1.00	1.00	1.00	1.00	1.00	1.00	1.00	1.00
	Black	1.17 (0.84–1.62)	1.08 (0.82–1.43)	0.85 (0.63–1.15)	1.05 (0.78–1.40)	0.89 (0.68–1.16)	1.01 (0.59–1.71)	1.16 (0.86–1.55)	0.85 (0.55–1.31)	1.34 (0.99–1.81)	0.75 (0.51–1.08)
	American Indian	1.64 (0.63–4.32)	1.20 (0.49–2.95)	0.91 (0.35–2.40)	0.68 (0.24–1.97)	1.40 (0.65–3.04)	2.00 (0.45–8.99)	1.11 (0.42–2.92)	1.46 (0.42–4.99)	1.16 (0.40–3.35)	0.56 (0.13–2.38)
	Asian	1.01 (0.64–1.59)	1.12 (0.77–1.62)	0.84 (0.55–1.28)	1.02 (0.68–1.53)	0.88 (0.60–1.29)	0.56 (0.20–1.57)	1.02 (0.67–1.57)	1.10 (0.58–2.08)	1.26 (0.82–19.4)	0.72 (0.42–1.23)
	Pacific Islander	2.26 (0.83–6.13)	0.85 (0.25–2.88)	2.02 (0.80–5.13)	0.85 (0.25–2.89)	1.07 (0.40–2.90)	^b^	0.66 (0.15–2.85)	1.52 (0.34–6.75)	1.32 (0.38–4.52)	1.36 (0.40–4.64)
	Other	1.04 (0.57–1.88)	0.86 (0.52–1.43)	1.12 (0.70–1.79)	0.48 (0.25–0.90)	1.02 (0.66–1.58)	1.83 (0.94–3.57)	0.55 (0.29–1.04)	0.82 (0.37–1.82)	0.90 (0.51–1.58)	0.57 (0.28–1.15)
	Multiple	0.96 (0.64–1.43)	1.40 (1.03–1.90)	0.98 (0.69–1.39)	0.80 (0.55–1.15)	1.00 (0.73–1.37)	1.34 (0.73–2.44)	1.39 (1.00–1.93)	1.49 (0.93–2.37)	1.52 (1.07–2.16)	1.13 (0.77–1.66)
Cigarette Smoking	Never Smoked	1.00	1.00	1.00	1.00	1.00	1.00	1.00	1.00	1.00	1.00
	Smoked but Quit	1.20 (0.93–1.56)	1.12 (0.90–1.39)	1.02 (0.80–1.28)	1.43 (1.15–1.79)	1.16 (0.95–1.43)	0.78 (0.50–1.21)	1.07 (0.84–1.36)	0.67 (0.46–0.97)	1.09 (0.84–1.42)	1.03 (0.78–1.35)
	Current Smoker	1.06 (0.81–1.39)	1.04 (0.82–1.31)	0.95 (0.74–1.22)	1.09 (0.86–1.38)	1.10 (0.89–1.36)	0.96 (0.62–1.48)	0.86 (0.66–1.12)	0.89 (0.62–1.29)	1.27 (0.98–1.65)	1.00 (0.75–1.33)
Smokeless Tobacco	Never Used	1.00	1.00	1.00	1.00	1.00	1.00	1.00	1.00	1.00	1.00
	Used but Quit	1.03 (0.73–1.46)	1.03 (0.76–1.38)	1.14 (0.83–1.56)	0.97 (0.71–1.32)	1.04 (0.79–1.37)	1.64 (1.00–2.71)	1.21 (0.88–1.67)	1.13 (0.71–1.81)	0.99 (0.70–1.41)	0.95 (0.65–1.39)
	Current User	0.99 (0.72–1.36)	1.09 (0.84–1.43)	1.09 (0.82–1.46)	1.14 (0.87–1.49)	1.18 (0.92–1.51)	1.47 (0.93–2.34)	1.28 (0.96–1.71)	1.39 (0.92–2.09)	1.13 (0.83–1.52)	1.33 (0.97–1.83)
Alcohol Use	0 mL/wk	1.00	1.00	1.00	1.00	1.00	1.00	1.00	1.00	1.00	1.00
	0.23–24.85 mL/wk	0.95 (0.73–1.24)	1.07 (0.86–1.33)	0.71 (0.56–0.86)	1.08 (0.86–1.37)	1.02 (0.82–1.26)	0.87 (0.56–1.35)	1.11 (0.87–1.41)	0.96 (0.67–1.38)	0.86 (0.66–1.13)	1.32 (1.00–1.74)
	24.86–71.69 mL/wk	1.03 (0.79–1.34)	1.19 (0.95–1.48)	0.84 (0.67–1.06)	0.99 (0.78–1.26)	1.17 (0.95–1.44)	0.87 (0.56–1.34)	1.23 (0.97–1.57)	1.17 (0.82–1.67)	1.03 (0.80–1.34)	1.22 (0.92–1.63)
	>71.69 mL/wk	0.98 (0.75–1.27)	0.96 (0.76–1.20)	0.80 (0.63–1.00)	1.31 (1.03–1.65)	1.13 (0.91–1.39)	0.88 (0.58–1.36)	1.16 (0.90–1.49)	1.16 (0.81–1.65)	1.03 (0.79–1.34)	1.37 (1.03–1.83)
Aerobic ExerciseDuration	≤90 min/wk	1.00	1.00	1.00	1.00	1.00	1.00	1.00	1.00	1.00	1.00
	91–180 min/wk	0.94 (0.74–1.19)	0.88 (0.71–1.09)	0.90 (0.72–1.12)	0.98 (0.79–1.22)	0.87 (0.71–1.05)	0.82 (0.53–1.27)	0.81 (0.64–1.02)	0.98 (0.70–1.37)	0.91 (0.70–1.17)	0.98 (0.75–1.27)
	181–300 min/wk	0.99 (0.76–1.28)	1.05 (0.84–1.30)	0.99 (0.79–1.25)	1.02 (0.80–1.29)	1.01 (0.82–1.24)	0.90 (0.58–1.41)	1.00 (0.78–1.27)	0.91 (0.63–1.30)	0.99 (0.76–1.30)	1.01 (0.76–1.33)
	>300 min/wk	1.03 (0.78–1.36)	1.04 (0.83–1.33)	0.94 (0.73–1.20)	1.04 (0.81–1.33)	1.02 (0.82–1.28)	0.73 (0.47–1.14)	1.04 (0.81–1.34)	0.79 (0.54–1.15)	0.97 (0.74–1.27)	0.95 (0.71–1.26)
Resistance Exercise Duration	≤45 min/wk	1.00	1.00	1.00	1.00	1.00	1.00	1.00	1.00	1.00	1.00
	46–135 min/wk	1.01 (0.80–1.28)	1.24 (1.00–1.54)	1.01 (0.80–1.27)	2.11 (1.66–2.67)	1.11 (0.91–1.36)	1.34 (0.73–2.45)	1.26 (0.99–1.61)	1.26 (0.85–1.87)	1.43 (1.07–1.89)	1.19 (0.90–1.58)
	136–300 min/wk	0.76 (0.59–0.98)	1.27 (1.02–1.59)	1.19 (0.94–1.49)	2.03 (1.59–2.59)	1.19 (0.96–1.42)	2.51 (1.45–4.33)	1.34 (1.04–1.71)	2.72 (1.90–3.90)	1.94 (1.47–2.55)	1.62 (1.23–2.14)
	>300 min/wk	0.59 (0.44–0.80)	1.68 (1.32–2.14)	1.61 (1.25–2.07)	1.98 (1.51–2.60)	1.13 (0.89–1.43)	6.69 (3.94–11.38)	1.89 (1.45–2.46)	3.37 (2.26–5.03)	3.04 (2.27–4.06)	2.31 (1.71–3.11)
Service Branch	Air Force	1.00	1.00	1.00	1.00	1.00	1.00	1.00	1.00	1.00	1.00
	Army	0.92 (0.73–1.16)	1.13 (0.93–1.36)	0.87 (0.71–1.06)	0.89 (0.72–1.09)	0.97 (0.81–1.16)	1.61 (1.10–2.34)	0.96 (0.78–1.18)	0.85 (0.63–1.15)	0.96 (0.77–1.20)	1.03 (0.82–1.30)
	Marine Corps	1.31 (0.97–1.78)	1.19 (0.91–1.57)	0.80 (0.59–1.09)	1.11 (0.84–1.47)	0.94 (0.72–1.23)	1.58 (0.96–2.59)	0.92 (0.68–1.26)	1.18 (0.77–1.81)	1.11 (0.81–1.52)	0.90 (0.63–1.29)
	Navy	1.15 (0.90–1.48)	0.98 (0.78–1.22)	1.04 (0.83–1.30)	1.15 (0.92–1.45)	1.16 (0.95–1.42)	0.94 (0.56–1.58)	0.88 (0.69–1.12)	0.81 (0.56–1.16)	1.04 (0.80–1.35)	0.98 (0.75–1.29)

^a^ All 10 models include gender, age, formal education, body mass Index, Hispanic ethnicity, race, smoking, smokeless tobacco use, alcohol use, weekly duration of aerobic and resistance training, and service branch. ^b^ This groups had no prohormone users. Abbreviations: HS = high school, AA = associate in arts degree, Grad = graduate.

## Data Availability

The datasets presented in this article are not publicly available because of US government restrictions but can be obtained from the author on reasonable request and development of a Data Sharing Agreement.

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
