# Peer review of "Longitudinal Changes in Dietary Supplement Use among United States Military Personnel: The US Military Dietary Supplement Use Study"

_nutrients, 2024, doi:10.3390/nu16152547_

Round 1
Reviewer 1 Report
Comments and Suggestions for Authors
I wanted to thank you for the opportunity to review this article. The topic is interesting however I would like to express my concerns regarding the presented manuscript.
Below you can kindly find my detailed comments.
The Authors have already published a lot of data coming from this cohort and similar topics, please explain and emphasize the novelty of the presented study and what is new compared to previous articles published by Authors.
Please elaborate on topic of supplements in adults to provide better introduction to the topic.
Line 61 - please provide the rationale of this study. Why the assessment of supplements use in this particular group was important?
I assuming that active-duty participants are healthy, however is there any data available regarding any co-morbidities?
Line 322 - are there any differences in use for other substances?
Line 360 - due to this differences I suggest indicating clearly that this is preliminary/pilot data
Please format citations according to the journal recommendations.
Author Response
Reply to Reviewer 1 (Reviewer’s comments preceded by an “R”; author’s replies preceded by an “A”)
R: I wanted to thank you for the opportunity to review this article. The topic is interesting however I would like to express my concerns regarding the presented manuscript. Below you can kindly find my detailed comments.
A: Thank you for taking the time to review this article and for your helpful comments.
R: The Authors have already published a lot of data coming from this cohort and similar topics, please explain and emphasize the novelty of the presented study and what is new compared to previous articles published by Authors.
A: As the reviewer notes, we have previously published a number of articles on the prevalence of dietary supplement (DS) use, adverse effects, and factors associated with DS use by military personnel. However, most of these studies have been cross-sectional, examining DS use at a single point in time. To the best of our knowledge, this study is unique in that it is the first in the civilian or military sector with an authentic longitudinal design that tracks the same group of participants over time. Previous studies have used repeated cross-sectional designs that examine different (but representative) groups of participants over time. Tracking the same participants over time allowed us to examine changes in the pattern of DS use. That is, it allows us to identify participants who continued use, discontinued use, and those who were previous non-users who started use in the follow-up phase. To the best of our knowledge, our study is also the first to examine factors associated with discontinuing DS use over an approximate 1 year follow-up time.
R: Please elaborate on topic of supplements in adults to provide better introduction to the topic.
A: In the new second paragraph of the Introduction we have now provided the prevalence of DS use in the civilian and military sectors and note the increase in use prevalence over time (Lines 44-52, clean, revised manuscript).
R: Line 61 - please provide the rationale of this study. Why the assessment of supplements use in this particular group was important?
A: As the reviewer suggested, we have now provided the importance of examining supplement use in military personnel in the new second paragraph of the Introduction (Lines 50-52 of clean, revised manuscript). In essence, the prevalence of use among military personnel are higher than in civilian samples and they use considerably different types of DSs, many that are often used by athletes. Also we note on Lines 53-58 (clean, revised manuscript) that previous studies examining temporal trends in DS use have used between-subject designs (i.e., repeated cross-sectional studies) that used different, but representative, cohorts. Our tracking of DS use in the same cohort over time allowed an examination of changes in patterns of DS use providing information on the proportion of individuals who continued use, discontinued use, and became new users in the follow-up period. This study was unique in that DS use was followed in the same participants over time.
R: I assuming that active-duty participants are healthy, however is there any data available regarding any co-morbidities?
A: As the reviewer notes, the US military only accepts healthy individuals for entry to service and there are many medical conditions that can disqualify individuals from service. Additionally, retention in service is largely dependent on continued health as a diagnosis of certain medical conditions can result in discharge from service. SMs frequently experience temporary illnesses and injuries that are treated by medical care providers both inside and outside the military care system, at no financial cost to the SM.
To directly answer the reviewer’s question, we have published separate detailed papers that address medical conditions in relation to DS use and we refer the reviewer to these papers: 1) Associations between clinically diagnosed medical conditions and dietary supplement use: The US Military Dietary Supplement Use Study. Public Health Nutrition 26(6): 1238-1253, 2023 (doi: 10.1017/S1368980023000095); 2) Association between chronic medical conditions and persistent dietary supplement use: the US Military Dietary Supplement Study. Nutrients. In Press
R: Line 322 - are there any differences in use for other substances?
A: At Line 334-336 in clean, revised manuscript, we state that “Higher odds of discontinuing use as training duration increased was apparent in almost all DS categories apart from combination products where odds of discontinuing use were similar across duration strata.” Thus, as training duration increased there were higher odds of discontinuing DSs in most categories except combination products. This paragraph (Lines 329-341) specifically addresses resistance training and as stated, almost all DS categories had a higher likelihood of discontinuation as the amount of resistance training increased.
R: Line 360 - due to this differences I suggest indicating clearly that this is preliminary/pilot data
A: Preliminary/pilot projects are generally designed to evaluate feasibility and potential problems and improve upon the study design prior to a larger full-scale research project. We conducted a pilot project prior to this larger investigation for this purpose; see reference #17. The limitations listed here (recall bias, social desirability, errors in self-observation, inadequate recall) are common in questionnaire studies of any type. We list these common shortcomings I the limitations section of the Discussion just to remind readers of typical drawbacks for studies of this type.
R: Please format citations according to the journal recommendations.
A: In the initially submitted version of the paper for peer review, the journal (Nutrients) has a “free format submission” that does not require a specific citation style. However, in the final version the journal format for references is required. We have now formatted the references according to journal style.
Reviewer 2 Report
Comments and Suggestions for Authors
This study investigated longitudinal patterns of change in DS use and factors associated with discontinuing DS use in a single group of active-duty United States military service members (SMs). Although this paper is good and has a great relevance for readers of nutrients, I have few comments prior the acceptance for publication.
Abstract:
To add more data (numbers) in the results section.
Introduction:
What is hypothesis of study?
Methods:
To add a clear design of study.
To add a number of people evaluated.
Results:
What is nutritional status rather than BMI of people included?
Discussion:
What are discussion regarding the use the supplements with guidelines of ACSM statement?
Author Response
Reply to Reviewer 2 (Reviewer’s comments preceded by an “R”; author’s replies preceded by an “A”)
R: This study investigated longitudinal patterns of change in DS use and factors associated with discontinuing DS use in a single group of active-duty United States military service members (SMs). Although this paper is good and has a great relevance for readers of nutrients, I have few comments prior the acceptance for publication.
A: Thank you for the time and effort you put into reviewing and commenting on our work.
R: Abstract: To add more data (numbers) in the results section.
A: As the reviewer suggests, we have added considerable numeric data to the abstract. These data are intended to show that although prevalence was similar in the baseline and follow-up periods, the pattern of use in different DS categories changed in the follow-up phase. That is in the follow-up, some participants continued use, others discontinued use, and others started use.
R: Introduction: What is hypothesis of study?
A: Our hypothesis was implied in the final paragraph of the Introduction, but not explicitly stated. We have now modified the Introduction to state the hypothesis in a more direct fashion. That is, we hypothesized that the pattern of DS use would change over time with some participants continuing use, some discontinuing use, and others becoming new users (Lines 60-61 revised, clean manuscript).
R: Methods: To add a clear design of study. To add a number of people evaluated.
A: Although the study design is mentioned in the title, we have added the word “longitudinal” to the first sentence in the Methods to indicate the study design (Line 66 of revised clean copy). The number of participants are provided in the Abstract (Line 6 of revised clean manuscript) and in the first sentence of the Results (Line 171 of revised, clean manuscript)
R: Results: What is nutritional status rather than BMI of people included?
A: We did not obtain data on the nutritional status of the participants. Because service members are surveyed so often the Army Research Institute (which approves all Department of Defense questionnaires) severely limited the number of questions we could include on our survey.
R: Discussion: What are discussion regarding the use the supplements with guidelines of ACSM statement?
A: ACSM does not have a position statement specifically addressing DSs. However, the position statement entitled “Nutrition and Athletic Performance” (Med Sci Sports Exerc 48(3):543, 2016) has a short section on “dietary supplements and ergogenic aids.” Being published in 2016, it is somewhat dated. The section on DSs in the position statement above covers use prevalence among athletes, motivations for use, and warns about safety and general lack of efficacy for most DSs. It notes the few DSs with positive efficacy. Since our study does not cover motivations for use, safety, or efficacy, the only overlap between the position statement and our article is use prevalence among athletes. We cover that in the Discussion (Lines 285-292 of the clean, revised manuscript).
Reviewer 3 Report
Comments and Suggestions for Authors
This great article examines the use of dietary supplements, which can indeed change significantly over time, even though the population studied is in a better physical shape than the average civilian is. The study is well worked out, the tables are very informative, and the results are well discussed. I have some remarks:
- Table 2: apart from race and ethnicity, is there also anything known on the participants' religion and the (non-)acceptance of supplements?
- Table 4: would the CMH test not be a better alternative to the McNemar test?
- Apparently, you have forgotten to include the two other service branches: the U.S. Space Force and the U.S. Coast Guard...
Comments on the Quality of English LanguageSome typo's throughout the text.
Author Response
Reply to Reviewer 3(Reviewer’s comments preceded by an “R”; author’s replies preceded by an “A”)
R: This great article examines the use of dietary supplements, which can indeed change significantly over time, even though the population studied is in a better physical shape than the average civilian is. The study is well worked out, the tables are very informative, and the results are well discussed. I have some remarks:
A: Thank you for taking time to review our article and for your comments.
R: Table 2: apart from race and ethnicity, is there also anything known on the participants' religion and the (non-)acceptance of supplements?
A: We did not ask participants about their religion or their acceptance/non acceptance of the use of DSs.
R: Table 4: would the CMH test not be a better alternative to the McNemar test?
A: We think that the McNemar Test was the best statistic to use in this case. The null hypothesis here was there would be no difference in the proportion of users/non-user in the baseline and follow-up periods. In Table 4 we showed that for most DSs categories (except joint health products and other DSs) we rejected the null hypothesis and accepted the alternative hypothesis that the proportion of DSs users differed in the baseline and follow-up periods. The Cochran Mantel Haenszel (CMH) Test examines the association between two categorical variables while controlling for the effect of a third variable. While it would have been interesting, this was not the intend of this portion of our analysis. Also, we would have had to run a large number of analyses controlling for various factors, perhaps those in Table 2.
R: Apparently, you have forgotten to include the two other service branches: the U.S. Space Force and the U.S. Coast Guard...
Yes, we only examined Army, Navy, Air Force, and Marines. We did not receive approval from the Department of Defense, Office of People Analytics, to survey the Coast Guard for this study. The Space Force is still part of the Air Force, with personnel “redesignated” to the Space Force, so likely we had some participants from the Space Force in our study. The Army, Navy, Air Force, Marines, Coast Guard and Space Force all are part of the Armed Services of the US. All services other than the Coast Guard are under the Department of Defense. The Coast Guard is under Homeland Security.